# Influence of the Milling Conditions of Aluminium Alloy 2017A on the Surface Roughness

**DOI:** 10.3390/ma15103626

**Published:** 2022-05-19

**Authors:** Lukasz Nowakowski, Marian Bartoszuk, Michal Skrzyniarz, Slawomir Blasiak, Dimka Vasileva

**Affiliations:** 1Department of Manufacturing Engineering and Metrology, Kielce University of Technology, 25-314 Kielce, Poland; lukasn@tu.kielce.pl (L.N.); mskrzyniarz@tu.kielce.pl (M.S.); sblasiak@tu.kielce.pl (S.B.); 2Department of Manufacturing Engineering and Materials Science, Opole University of Technology, 45-758 Opole, Poland; 3Department of Mechanical Engineering and Machine Tools, Technical University of Varna, 9010 Varna, Bulgaria; d.vasileva@tu-varna.bg

**Keywords:** aluminium, face milling, surface roughness, microgeometric surface structure, relative displacement, minimum thickness of cut layer

## Abstract

The article presents the results and process analysis of the face milling of aluminium alloy 2017A with the CoroMill 490 tool on an AVIA VMC 800 vertical milling centre. The study analysed the effects of the cutting speed, the feed rate, the actual number of teeth involved in the process, the minimum thickness of the cut layer (*h_min_*), and the relative displacement in the tool-workpiece system *D(ξ)* on the surface roughness parameter *Ra*. To measure relative displacement, an original bench was used with an XL-80 laser interferometer. The analysis of relative displacement and surface roughness allowed these factors to be correlated with each other. The purpose of this article is to determine the stable operating ranges of the CoroMill 490-050Q22-08M milling head with respect to the value of the generated relative displacement w during the face-milling process and to determine its influence on surface roughness. The research methodology presented in this paper and the cutting tests carried out allowed the determination of the optimum operating parameters of the CoroMill 490-050Q22-08M tool during the face milling of aluminium alloy 2017A, which are *v_c_* 300 m/m and *f_z_*—0.14 mm/tooth. Working with the defined cutting parameters allows all the cutting inserts in the tool body to be involved in shaping the geometrical structure of the surface, while maintaining a low vibration level *D(ξ)* > 1 µm, a low value of the parameter *h_min_* > 1.5 µm, and the desired value of the parameter *Ra* > 0.2 µm

## 1. Introduction

Despite the development of various manufacturing processes, machining is still the primary method of producing machine parts. As is well known, milling processes account for a large proportion of machining. Such machining is carried out using milling cutters and high cutting parameter values. The continuous modification of cutting tool geometries and the development of tool materials and coatings mean that surfaces machined by end milling are usually not subject to further finishing operations. Thus, surfaces treated in this way gain their final surface microgeometry and surface layer properties at this very stage of processing.

Unambiguously defining and describing the milling process is very difficult, as it is accompanied by many disturbing factors [1]. In practice, carbide inserts in the milling head body are not perfectly aligned, which, in combination with the feed motion, affects the feed rate per tooth, the actual length of the cutting edges involved in material removal [2], and the microgeometry of the generated surface [3]. As demonstrated in the study by Arizmendi et al. [4], surface microgeometry can be affected even by inaccurate positioning of the cutting tool. They demonstrated that the non-homogeneous roughness bands produced on the machined surface in circumferential milling depend on the errors of the tool setting, its geometrical tolerance, and the feed value. The milling process is also accompanied by relative displacements of the tool-workpiece system, which are an unfavourable and undesirable phenomenon. They decrease the dimensional and shape accuracy of machined parts and the quality of machined surfaces, which causes a decrease in the profitability of the manufacturing process, as observed in the work of Skrzyniarz et al. [5]. Displacements generated in the workpiece-tool system also reduce tool and machine component life. Furthermore, they are an obstacle to increasing the efficiency of the workpiece material removal process and significantly affect the geometric structure of the machined surface [6]. The influence of these displacements on the quality of machined surfaces was studied by Fei et al. [7]. The results of their research indicated that deformation during machining is the main cause of poor surface quality of a workpiece, especially a flexible workpiece. Li et al. [8] pointed out the great importance of the workpiece clamping method. Improper clamping can seriously affect the final quality of the machined surface, especially the accuracy of the shape. This issue is particularly important when machining thin-walled components, as highlighted in the study by Wu et al. [9]. 

On the other hand, Yang and Liu [10] showed that in addition to workpiece deflection, tool deflection and tool run-out have a significant influence on the profile of the machined surface. In turn, Morelli et al. [11] drew attention to the dynamic stiffness of the tool and indicated that changing this parameter contributes to surface errors in milling operations. It should be noted, however, that by treating the tool setting as an adjustable parameter, it is possible to influence the final machining effect. Liu et al. [12] showed that inclining milling tools at a suitable angle can help to significantly suppress vibration during machining and improve the quality of the machined surface. The importance of vibration in the milling process is also indicated by the results of research conducted by Wang et al. [13]. They proved that changing the dynamic stiffness of the machine tool changes the vibration frequency of the system, which then leads to different surface quality. 

Tool vibration can be influenced by optimising the machining parameters. Yan et al. [14] showed that when tool vibration is kept to a minimum, the roughness of the machined surface depends mostly on the feed value and least on the depth of cut. The influence of machining parameters on the roughness of the machined surface was studied by Wang et al. [15]. In order to improve the quality of the machined surface, they undertook to optimise the parameters of the cutting process. By varying the spindle speed, the feed rate, the depth of cut, and the cutting distance, they investigated how these parameters affect surface topography. The study was conducted for a micro-cutting process. In micro-cutting, we often have to deal with the value of the smallest removable allowance under given machining conditions. This value is defined by the parameter of the minimum thickness of the machined layer *h_min_* [16,17]. Instead of cutting, the process of surface burnishing takes place. Oliveira et al. [18] found that the numerical value of the *h_min_* parameter varies practically from 1/4 to 1/3 of the value of the rounding radius of the cutting edge of the cutting tooth, regardless of the workpiece material, the tool geometry, the machining process, or the method of measurement used. Skrzyniarz [19] presented a method for determining the *h_min_* parameter by analysing the profile of the produced surface.

In summary, the factors affecting the roughness of the machined surface can be divided into four groups: machining parameters (eight variables), tool properties (nine variables), workpiece material (six variables), and phenomena accompanying the cutting process (eight variables). This division is shown in detail in Figure 1.

In practice, it is common to first select the machining parameters suggested by the tool manufacturer and then optimise their values in light of the process characteristics. The process of optimising the cutting parameters is often supported by analytical [20,21] or simulation models [22,23] that are used to predict the surface roughness of machined surfaces. However, these models are not perfect. Therefore, they are constantly under development and supplemented with new data. For example, Zawada-Michałowska et al. [24] pointed out that machining accuracy is influenced by the machining strategy and the direction of tool guidance, which was also described by Dzyura et al. [25] and Chuchala et al. [26]. This proves that milling in the direction perpendicular to the rolling direction (pre-milling) causes greater deformation than milling in the parallel direction.

Therefore, it is easy to see that there are many factors that can influence the surface forming process during machining. Accordingly, intelligent algorithms are being introduced into the decision-making processes in machining. A genetic algorithm, namely particle swarm optimisation, and other algorithms are used to optimise process parameters in order to achieve the ideal selection of machining parameters [27,28,29]. However, the data describing the factors influencing surface topography are in a closed set; therefore, constant replenishment and multi-threaded analysis of interrelationships is necessary.

In industrial practice, only the basic parameters of the machining process, such as cutting speed *v_c_*, depth of cut *a_p_*, and feed rate *f_z_*, are selected to obtain a machined surface with a specified roughness. No other factors affecting surface microgeometries are considered. This article presents a new methodology for selecting parameters of the milling process, taking into account not only the basic process parameters, but also relative tool displacement *D*, the minimum thickness of the machined layer *h_min_*, or the number of blades involved in cutting. The research was conducted on a special measuring machine, the construction details of which are protected by patent P.393156, “Combined machine tool and laser system for relative displacement measurement.” The research was aimed at determining the influence of selected cutting parameters (feed per tooth *f_z_* and cutting speed *v_c_*) on the process of face milling of the surface of aluminium alloy 2017A. Among other things, the authors analysed the influence of the actual number of teeth involved in the cutting process, the effect of relative displacements in the tool-workpiece system, and the value of the minimum thickness of the machinable layer on the geometric structure of the machined surface.

## 2. Materials and Methods

The material used for face milling was aluminium alloy 2017A. This alloy has good strength properties and high tensile and fatigue strength. It is used in the manufacture of components for aircrafts, military equipment, machine building parts, and automotive components. The chemical composition of the workpiece material is shown in Table 1, while its basic mechanical properties are shown in Table 2. The material properties were certificated by the material supplier.

A CoroMill 490-050Q22-08M head (Sandvik Coromant, Sandviken, Sweden), shown in Figure 2, was used as the cutting tool in this study. The tool has a normal insert seat pitch, with the tool cutting edge angle of 90°, which results in a predominantly radial force in the feed direction, so that no excessive axial pressure is exerted on the machined surface, which is important in milling workpieces with a weak structure or thin walls, and in cases of unstable clamping. The Coro Mill 490 milling head is equipped with inserts, type 490-08T308M-PL-1030. These are right-hand inserts that are 8 mm wide and 3.97 mm thick. The corner radius was 0.8 mm. The inserts are designed for light processing.

Inserts designated 490-08T308M-PL-1030 (Sandvik Coromant, Sandviken, Sweden) made of sintered carbide GC1030 (HC)—P30, with PVD TiAlN protective coating, were selected as cutting tooths [30].

The tests were conducted on specimens with an octagonal cross-section (Figure 3). This shape of the specimens allowed them to be securely clamped in the machine vice and allowed the workpiece material to be used economically. A precision machine vice type 6620 (Bison-Bial, Bielsk Podlaski, Poland) was used as the clamping system during all cutting tests. The average measured value of the Ra parameter of the working surface roughness of the specimens before the cutting tests was 4.5 µm. Other values of selected geometric structure parameters of the sample surface are shown in in Chapter 3.

Prior to the experimental tests, the alignment of the replaceable cutting inserts in the milling head and the error of their attachment in the tool body slots were measured. This made it possible to determine the actual number of cutting teeth involved in shaping the geometric structure of the surface during the milling process. The measurements were carried out using the Kalimat-C (Kelch GmbH, Weinstadt, Germany) device. A geometric model of the tool was then developed, which allowed the number of teeth involved in the cutting process to be determined for specific feed rate ranges.

During the milling process, the relative displacement signal in the tool-workpiece system was recorded. These vibrations were measured using a test bench equipped with an XL-80 interferometric laser system (Renishaw plc, New Mills Wotton-under-Edge Gloucestershire, UK) and an accurate measurement track (Figure 4).

Surface roughness and topography measurements for all samples were carried out after testing, using measuring instruments from Taylor Hobson: Talysurf CCI – Lite Non–contact 3D Profiler (Taylor-Hobson Ltd., Leicester, UK). The surface was measured using an eyepiece with a magnification of 20 times, which allowed an area of 0.8 mm × 0.8 mm to be measured. The measured surface consisted of 1024 × 1024 measurement points from which, after applying a Gaussian cut-off filter 0.8, the 2D profiles of the surface were determined. In measuring 2D parameters, information was collected on *Rp*—the maximum peak height of the roughness profile; *Rv*—the maximum valley depth of the roughness profile; *Rz*—the maximum height of the roughness profile; *Rc*—the average height of the roughness profile elements; *Rt*—the total height of the roughness profile; *Ra*—the arithmetic mean deviation of the roughness profile; *Rq*—the mean square deviation of the roughness profile; *RSm*—the average width of the roughness profile elements; and *Rdq*—the mean-square slope of the roughness profile. The analyses also estimated the value of their standard deviation. The 3D parameter values of the tested surfaces were also determined: *Sq*—the mean square height of the surface; *Sp*—the maximum peak height of the surface; *Sv*—the maximum height of the surface indentation; *Sz*—the maximum height of the surface; and *Sa*—the mean arithmetic height of the surface. All the aforementioned roughness parameters were determined from the averaged profile, understood as the arithmetic mean profile determined for the measured microgeometry field of the analysed surface. As a result, an average profile was calculated from several hundred individual roughness profiles.

Only on the basis of the information gathered in this way were the analyses conducted of the influence of the studied quantities on the roughness and topography of the surfaces obtained in the face-milling process of aluminium alloy 2017A.

Cutting tests were carried out on a VMC 800 vertical machining centre (AVIA, Warsaw, Poland) without the use of a cooling lubricant. The cutting process parameters were adopted based on previous research. The process of cylindrical milling of the sample surface was carried out at a constant cutting speed *v_c_* = 300 m/min with variable feed per tooth in the range of 0.02–0.22 mm/tooth (graded every 0.02) and at a constant feed per tooth *f_z_* = 0.1 mm/tooth, for which the cutting speed was changed in the range of 200–400 m/min (graded every 20 m/min). During the milling process, the relative displacement signal in the tool-workpiece system was recorded using the test bench shown in Figure 4.

Aluminium 2017A is a low hardness alloy with good machinability. Therefore, no signs of cutting tool wear were observed in the experiments conducted. In addition, new cemented carbide inserts were used for testing each time. For this reason, the effect of cutting blade wear on the microgeometry of the surfaces studied is omitted in this paper.

## 3. Results and Discussion

Inaccuracies in the clamping of cutting inserts in the body of the milling head were taken as one of the obvious factors affecting the roughness of the machined surface. Therefore, analyses of these inaccuracies were carried out in the first instance. The results obtained are shown in Table 3. Figure 5 shows a diagram of the measurement of the inaccuracy of the clamping of the cutting inserts in the body of the milling head. The results of the measurements are shown as the deviation of the position of the measured insert from the adopted reference insert. The reference insert is understood to be the cutting insert furthest from the seat. The measurements show that the furthest-reaching insert in the axial direction of the tool is insert 1 (Δ*L* = 0), while for the radial dimension it is insert 4 (Δ*R* = 0). Fifty measurements were taken of the alignment of each insert by determining the average axial and radial alignment error.

The analysis of the presented results confirms that the value of the feed per tooth and errors in the positioning of the inserts in the tool body can significantly affect the material removal process. At a feed rate of 0.04 mm/tooth, only three inserts out of five clamped in the tool body are involved in the surface geometric structure shaping process. When the feed rate exceeds 0.12 mm/tooth, all the teeth are involved in the material removal process, as the feed rate per tooth is higher than the largest value of the radial and axial insert clamping error in the tool body. For measurements of inaccuracies in the mounting of cutting inserts, an estimation of measurement uncertainty was performed, based on the guidelines contained in Polish standards PN-EN ISO 14253-2:2011E and PN-EN ISO 5436-1:2002: accuracy—4.5 μm; repeatability of length measurement L—1.3 μm; repeatability of radius measurement R—0.3 μm; composite standard uncertainty—4.6 μm; and expanded uncertainty (k = 2)—9.2 μm.

Another equally important factor determining the microgeometries of the machined surface is the relative displacement *D* in the tool-workpiece system. Figure 6 shows an example plot of the relative displacement signal *D* for the machining process of aluminium alloy 2017A with the following parameters: *v_c_* = 200 m/min, *f_z_* = 0.1 mm/tooth, depth of cut *a_p_* = 1 mm, milling width ae = 24 mm, and number of teeth involved in the machining allowance removal process *z* = 4. For the case studied, the value of relative displacement *D* did not exceed 2.2 µm during the entire test. The graph shows a certain cyclicality in the variation of the waveform *D(t)*. This is due to inaccuracies in the clamping of the cutting inserts in the milling head body, which translate into an uneven load on the individual teeth in the milling head, and to the speed of the tool. This means that one cycle of chart change corresponds to one rotation of the tool. The tool head for a cutting speed of 200 m/min performed 1273 rpm. Figure 6 shows characteristic peaks, whose frequency of occurrence coincides with the time of one full revolution of the tool head. The maximum values in the diagram correspond to the work of insert no. 1, which is seen to remove the largest cross-sectional area of the material being cut. Then, there is a decrease in displacement due to the displacement of the insert in the tool body, which causes a decrease in the cross-sectional area of the cut layer and consequently a lower load on it. The minimum values in the graph correspond to the work of insert no. 3, which was not involved in the material removal process. In the analyses performed, the value of the relative displacements in the tool-workpiece system is represented by the standard deviation parameter D(ξ) of these displacements. For the measurement of the relative displacements *D* in the tool-workpiece system, an estimation of the measurement uncertainty was performed based on the abovementioned standards: indication error—0.003 μm; repeatability—0.02 μm; resolution—0.0002 μm; composite standard uncertainty—0.02 μm; and expanded uncertainty (k = 2)—0.04 μm.

Figure 7 and Figure 8 show the effect of velocity *v_c_* and feed rate *f_z_* on the standard deviation of relative displacements in the tool-workpiece system *D(ξ)*. When considering the effect of cutting speed on the standard deviation of relative displacements (Figure 7), there are two characteristic maximum values of the parameter *D(ξ)*: for *v_c_* equal to 260 m/min and 360 m/min, the values of the parameter *D(ξ)* were up to, respectively: 2.69 µm and 2.36 µm. The aforementioned characteristic points define the favourable working range of the CoroMill 490 tool and the AVIA VMC 800 milling centre. By analysing the graph in Figure 7, it can be concluded that the most favourable values of the cutting speed, due to the smallest values of the standard deviation of the relative displacements, are for the speeds 200–240 and 280 m/min, for which the parameter *D(ξ)* oscillates around 1 µm.

Figure 7 also summarises the changes in *h_min_*, *D(ξ)* and the parameter *Ra* as a function of cutting speed. Analysing the course of changes of the *Ra* parameter, it can be seen that the use of low cutting speeds causes deterioration of the surface roughness and the need for greater depths of cut, as the *h_min_* parameter reaches its highest values in this interval. Analysing the course of changes in the minimum thickness of the machined layer *h_min_*, it can be observed that in the initial phase, an increase in the cutting speed *v_c_* caused a significant decrease in the value of the *h_min_* parameter. A further increase in the value of the *h_min_* parameter was observed when the cutting speed exceeded 220 m/min. This upward trend continued, up to a cutting speed of 260 m/min. Further increases in the cutting speed resulted in alternating decreases and increases in *h_min_*, until *v_c_* = 380 m/min, where a decrease in *h_min_* occurred. This phenomenon can be explained by the fact that the cutting speed has no significant effect on the values of the parameter *h_min_*, and its maximum value is correlated with the highest value of the parameter *Ra* of the surface roughness.

For the measurement of the minimum thickness of the machined layer, an estimation of measurement uncertainty was performed, based on the guidelines provided in the standards: measurement accuracy—0.014 μm; repeatability—0.01 μm; composite standard uncertainty—0.02 μm; and expanded uncertainty (k = 2)—0.04 μm.

The analysis of the results presented in Figure 8 clearly shows the adverse effect of increasing the feed rate on the value of the parameter *D(ξ)*. Increasing the feed rate per tooth resulted in a systematic increase in the standard deviation of relative displacement in the tool-workpiece system, from a minimum value of 0.60 µm for a feed rate of 0.02 mm/tooth to a maximum value of 1.90 µm for a feed rate of 0.18 mm/tooth. This phenomenon is caused by an increase in the cross-sectional area of the cut layer, which translates into a higher load on the individual insert during operation.

In the presented list of results of measurements carried out with a variable feed per cutting insert, the vertical lines indicate the intervals that represent the number of teeth involved in the process of shaping the geometric structure of the surface. The characteristic point on the diagram is the feed rate value of 0.12 mm/tooth, as all cutting inserts take part in the surface shaping process. This phenomenon is reflected in the graph by the decrease in the value of the standard deviation of relative displacements, for a feed rate of 0.14 mm/tooth in relation to a feed rate of 0.12 mm/tooth. This translates into the involvement of the fifth cutting insert in material removal and a change in the relative displacement characteristics generated in the tool-workpiece system.

Although for low feed rates small values of the *h_min_* parameter were determined, and the lowest relative displacements in the tool-workpiece system for the entire range of feed rates were studied, the *Ra* parameter of the surface roughness reached its highest value at 2.5 µm. The low feed rate per cutting edge, combined with faulty attachment of the cutting inserts in the tool body, meant that only three of the five cutting inserts were involved in the material removal process. During the cutting process with the 490-08T308M-PL-1030 insert, the theoretical value of the chip cross-sectional area for a feed rate of 0.02 mm/tooth and a cutting depth ap = 0.2mm was determined to be 0.004 mm^2^. After taking into account the errors of clamping the inserts in the head body, there is an asymmetry in the cross-sectional areas of the material removed by the individual inserts, because insert 1 removed 0.0108 mm^2^, insert 2 removed 0.005 mm^2^, insert 4 removed 0.0042 mm^2^, and inserts 3 and 5 were not involved in the process of shaping the geometric structure of the surface. In this case, insert 2, which removed very little material, caused an increase in the Ra value through a phenomenon known as material furrowing, due to the fact that the *h_min_* parameter was not exceeded.

Analysing the effect of the feed rate per cutting insert *f_z_* on the parameter of minimum thickness of the machined layer *h_min_*, it was found that in the initial phase, the value of the parameter of minimum thickness of the machined layer gradually increased as the feed rate increased (0.02–0.08 mm/tooth). For the feed rate range of 0.08–0.16 mm/tooth, the *h_min_* parameter oscillated around the value of 2 µm, after which, with a further increase in the cutting speed, only a gradual decrease in the value of the *h_min_* parameter was observed. An increase in the cutting speed increased the dynamics of the material separation process, causing a decrease in the trend of the h*_min_* parameter value. Aluminium alloys, especially at low cutting speeds, have a tendency to “smear” and build up, which compromises the quality of the machined surface.

The analysis of the geometric structure of the face-milled surfaces was carried out in two stages: in the first stage, the influence of the feed was analysed, while in the second stage, the focus was on the influence of the cutting speed. The developed measurement results are presented in the form of an atlas of geometric structures of surfaces after machining with different cutting parameters. The results are summarised in the form of tables and graphs, including cutting parameters, isometric images of the surface, 3D surface roughness parameters, surface profiles, and 2D surface roughness parameters.

Figure 9 shows isometric images of the measured surfaces of samples made of aluminium alloy 2017A. Initially, at a very low feed rate of 0.02 mm/tooth, the profile was random, with flat sections divided by 1.5 μm deep scratches (Figure 9a). As the feed rate per tooth increased to 0.1 mm/tooth (Figure 9e), the cracks appeared less and less frequently and their depth decreased slightly to about 1 μm. For feed rates *f_z_* = 0.1 ÷ 0.18 mm/tooth (Figure 9e–i), it was observed that the profile gradually changed its character from short-wavelength to long-wavelength, where the surface alternated from one with scrapes to one without scrapes. A further increase in the feed rate per tooth did not change the surface roughness, but increased the surface waviness. For feed rates in the range *f_z_* = 0.18 ÷ 0.22 mm/tooth (Figure 9i–k), traces of radial multidirectional machining appear on the isomeric image of the surface, which is the result of the milling head cutting anteriorly and posteriorly. The parameter maximum peak height of the roughness profile *Rp* in the feed rate range 0.02 ÷ 0.06 mm/tooth (Figure 9a–c), showed a decreasing trend, from a maximum value of 0.818 μm to 0.552 μm. A further systematic increase in the feed rate resulted from 0.08 mm/tooth (Figure 9d) in a slight increase in the value of the *Rp* parameter to 0.589 μm, followed by a decrease to a minimum value of 0.53 μm. As the feed per cutting insert increased, there was a slight increase in the *Rp* parameter to 0.643 μm, followed by a stabilisation at 0.6 μm. The course of changes in the *Rv* parameter, after an initial decrease from a maximum of 1.85 μm to 0.811 at a feed rate of 0.06 mm/tooth, stabilised at an average level of about 0.75 μm, reaching a minimum of 0.663 μm at a feed rate of 0.18 mm/tooth (Figure 9i).

The maximum height parameter of the roughness profile *Rz* has the same waveform as the parameter *Rv*. The parameter of the average height of the elements of the roughness profile, with an increase in feed rate, like the previous parameters, showed a decreasing trend to the value of feed rate 0.06 mm/tooth (Figure 9c); after that value was exceeded, the parameter oscillated around the average value of 0.5 μm. The parameter of total height of the roughness profile *Rt* showed a downward trend from its maximum value at a feed rate of 0.02 mm/tooth (Figure 9a) to its minimum of 1.871 μm at a feed rate of 0.1 mm/tooth (Figure 9e). For the further range of feed rates tested, the parameter *Rt* showed a small but systematic increasing trend up to a feed rate of 0.2 mm/rev (Figure 9j), after which it decreased to a value of 2.033 μm. Raw surface before machining process with variable feed rate is shown in Figure 9l. The numerical values of the roughness parameters tested and the standard deviation values calculated for them are summarised in Table 4.

The extracted parameter values of the arithmetic mean deviation of the roughness profile *Ra* are shown in Figure 10. In the initial phase of the increase in the feed rate, the *Ra* parameter showed an intensive downward trend until reaching *f_z_* = 0.06 mm/tooth. (The use of low feed rates caused the honing edge of the cutting insert to shape the same surface several times, which resulted in an increase in the value of parameter *Ra*). Beyond this value, the *Ra* parameter stabilised at 0.15 μm. In the feed rate range of 0.1 ÷ 0.14 mm/tooth, *Ra* increased to a value of 0.193 μm, after which the trend reversed to a downward trend that persisted up to a feed rate of 0.18 mm/tooth. For a feed rate of 0.2 mm/tooth, the Ra parameter increased its value once again, only to fall again. The characteristics of the behaviour of the *Rq* parameter were identical to those of the *Ra* parameter over the entire range of the feed rate tested (Table 4). The average width of the *RSm* roughness profile elements, with an increase in feed per tooth, began to increase its value after a slight decrease at the beginning, systematically, reaching a maximum of 0.032 mm for feeds of 0.14 mm/tooth. For a feed rate of 0.16 mm/tooth, there was a slight decrease in *RSm* to 0.027 mm, followed by an increasing trend. The lowest *RSm* value of 0.011 mm was measured at a feed rate of 0.04 mm/tooth. Increasing the feed rate had a large effect on the mean square slope of the *Rdq* roughness profile only for small feed rates in the range 0.02 ÷ 0.1 mm/tooth, where the greatest change in the *Rdq* parameter occurred from 14.244° ÷ 4.507°. A further increase in feed rate per tooth did not cause any significant changes in parameter *Rdq* (Table 4).

Analysing the results obtained for a feed rate of 0.12 mm/min located in the middle of the measuring range, it can be seen that the main parameters describing the surface roughness were reduced by approximately 90%. Examples of changes are, respectively, *Ra*—96.3%; *Rt*—87.8%; *Rp*—93.3%; and *Rv*—91.5%. Only the mean square profile elevation *Rdq* decreased, by 39.1%. Similarly, the profile height parameters changed to the basic parameters: e.g., for *Sp* there was a decrease of about 91.1% and for *Sv* there was a decrease of about 80.0%. Comparing the results in Table 4 and Figure 8, it is easy to see that as the number of blades involved increases, the surface roughness decreases. In the case in question, five cutting edges are involved in the cutting process, which has a direct impact on the improved roughness parameters. It can also be assumed that the roughness is influenced by the cutting parameters themselves, because as the feed rate increases, the cutting temperature changes, which affects the material separation mechanism.

The following diagrams and tables show a summary of the isometric images and numerical values of the results obtained for machining with a variable cutting speed at a constant feed per cutting insert of *f_z_* = 0.1 mm/tooth and a constant depth of cut of *a_p_* = 0.2 mm.

Isometric images of the measured surfaces obtained by face milling of samples made of aluminium alloy 2017A are shown in Figure 11. For a low cutting speed of 200 m/min (Figure 11a), the measured surface showed material loss in the form of cracks about 4 μm deep, which were located at a distance corresponding to the feed per cutting insert, which translated into a random character of the profile. For a cutting speed of 220 m/min (Figure 11b), a change in the shape of the profile to longitudinal was observed, which oscillated within ±1 μm. For a cutting speed of 240 m/min (Figure 11c), the amplitude of the profile oscillation remained the same, but its period lengthened to a value equal to the feed per revolution of 0.5 mm. At a cutting speed of 260 m/min (Figure 11d), the surface profile was characterised by sharp peaks, with an inter-peak value of about 1.5 μm. A further increase in the cutting speed resulted in sporadic material loss in the form of tears and craters with a depth of 1 ÷ 2 μm. In the case of the aluminium alloy 2017A tested, the characteristics of the course of change of the parameters *Rp*, *Rv*, *Rz Rc*, *Ra*, *Rq*, and *Rdq* with an increase in the cutting speed showed an almost identical course throughout the test range. The numerical values of the roughness parameters tested and the standard deviation values calculated for them are summarised in Table 5.

Analysing the results obtained for a cutting speed of 300 m/min located in the middle of the measuring range, it can be seen that the main parameters describing the surface roughness were reduced by approximately 90%. Examples of changes are, respectively, *Ra*—95.7%; *Rt*—87.5%; *Rp*—92.8%; and *Rv*—90.9%. Only the mean square profile elevation *Rdq* decreased, by 33.8%. Similarly, the profile height parameters changed to the basic parameters: e.g., for *Sp* there was a decrease of about 89.6% and for *Sv* there was a decrease of about 79.6%.

A description of an exemplary course of changes in the characteristics of the above-mentioned parameters as a function of the increasing cutting speed is described in detail on the example of the *Ra* parameter (Figure 12). In the initial phase of increasing the cutting speed from 200 to 220 m/min, the *Ra* parameter decreased significantly from a maximum value of 0.76 μm to 0.29 μm. In the next range of cutting speeds, i.e., 220 ÷ 260 m/min, the *Ra* value decreased again, less intensively to 0.18 μm. Further, in the range of cutting speeds 260 ÷ 400 m/min (Figure 11d–k), the *Ra* parameter alternately increased and decreased its value by 0.03 ÷ 0.05 μm, oscillating around the value of 0.2 μm. The *Rt* parameter, on the other hand, recorded its maximum at a very low cutting speed of 4.78 μm and decreased to 2.53 μm when the cutting speed was increased by 20 m/min (Table 5). A further increase in the cutting speed to 240 m/min resulted in a slight increase in the parameter *Rt* to 2.67 μm, which then decreased its value to 2.24 μm. Raw surface before machining process with variable cutting speed is shown in Figure 11l.

At a cutting speed of 280 m/min, the *Rt* value jumped to 3.27 μm, recovered to 2.26 μm after increasing the cutting speed, and stabilised at this level up to a cutting speed of 340 m/min (Table 5). At a cutting speed of 360 m/min, there was a marked increase in the value of the parameter to 2.92 μm, which decreased and increased its value by an average of around 0.85 μm in the further range of cutting speeds up to 400 m/min.

An increase in the cutting speed in the range 200 ÷ 400 m/min caused changes in the *Rdq* parameter, of the same character as for the *Rt* parameter (Table 5).

## 4. Conclusions

On the basis of our research, it can be concluded that:The face milling of cylindrical aluminium alloy 2017A with the CoroMill 490-050Q22-08M head with *v_c_* 300 m/m and −0.14 mm/tooth allows the involvement of all cutting inserts in the tool body in the process of shaping the geometric structure of the surface, while maintaining a low vibration level of *D(ξ)* > 1 µm, a low value of the parameter *h_min_* of >1.5 µm, and the desired value of the parameter, *Ra* > 0.2 µm.A cutting speed of 200 m/min and low feed rates per cutting insert results in a deterioration of a surface geometric structure for aluminium alloy 2017A by increasing surface height parameters.Characteristic cutting speed ranges of 200–240 m/min and 280 m/min were determined where the standard deviation of relative displacement had the lowest value, *D(ξ)* = 0.98 to 1.07 µm.The value of the standard deviation of the relative displacements shows an increasing trend as the feed per tooth increases. However, there are characteristic boundaries responsible for increasing the proportion of the inserts involved in the material removal process that reverse or significantly reduce this trend.The characteristic intervals for the cutting tool in which the values of relative displacements are highest (*v_c_* = 280 m/min, *f_z_* = 0.18 mm/tooth) were determined.The characteristics of the relative displacement signal in the tool-workpiece system are correlated with the cross-sectional area of the material removed by the individual cutting inserts. Using the displacement signal, it is possible to diagnose the number of inserts involved in the cutting process.Analysis of insert clamping errors in the tool body enables determination of optimum feed rates at which all cutting inserts will participate in shaping the geometric structure of the surface. At a feed rate of 0.04 mm/blade, only three inserts out of five are involved in the cutting process. It is only when the feed rate exceeds 0.12 mm/blade that all the blades are involved in the material removal process, which is due to the machining accuracy of the tool body and the manufacturing errors of the cutting inserts.The development of a geometrical model of the tool made it possible to estimate the load of the individual cutting inserts and the values of the actual feeds per blade, to determine the instantaneous depths of cut, and to calculate the cross-sectional area of the material removed for each insert.The methodology presented in this paper allows optimal selection of cutting parameters for the milling process, based on information concerning the relative displacement of the tool-workpiece system, the minimum thickness of the cut layer, the number of cutting edges involved in the process of shaping the geometric structure of the surface, and selected parameters of the geometric structure of the surface.The direction of future research will be concerned with the simplification of the relative displacement measurement procedure in the tool-workpiece system, which currently requires the use of a laser interferometer and a process of filtering and processing the signal, hindering the expansion of this method in industrial practice.The test bench and measurement methodologies used in this study are universal and can be applied to any milling machine, regardless of its kinematic system and control method.

## Figures and Tables

**Figure 1 materials-15-03626-f001:**
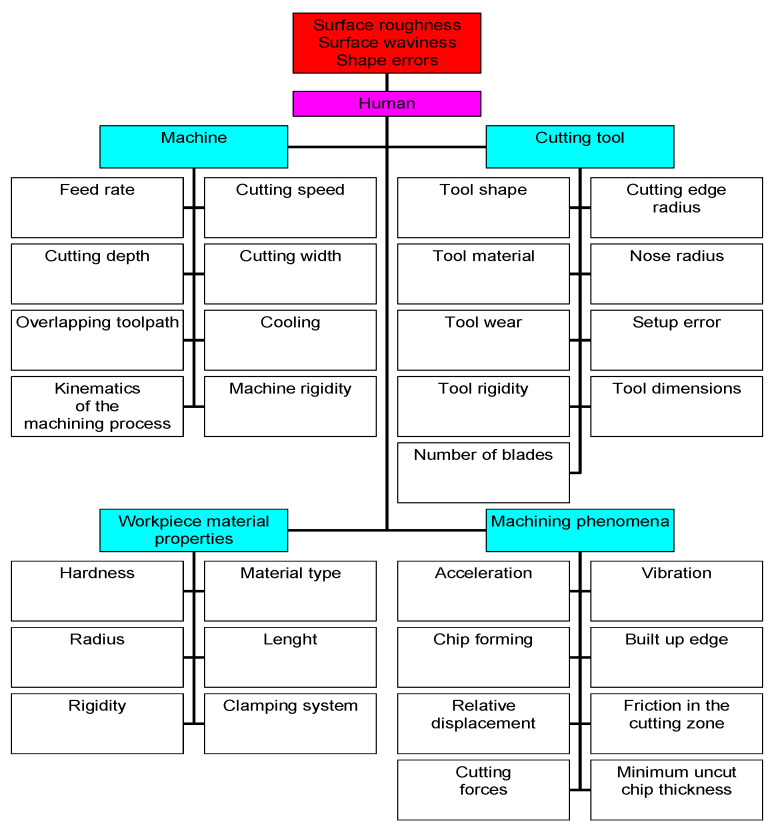
Factors affecting the geometric structure of the surface during the machining process.

**Figure 2 materials-15-03626-f002:**
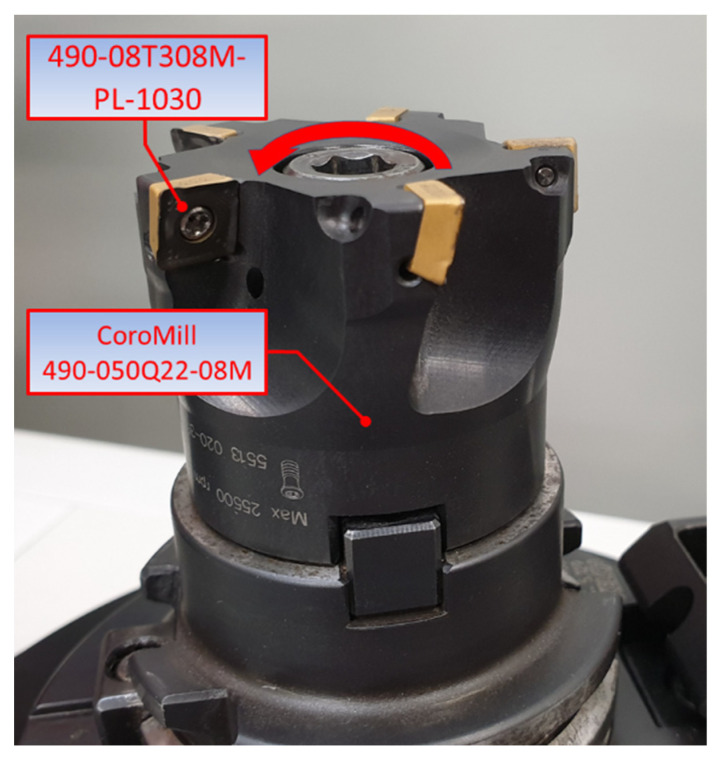
View of the CoroMill 490-050Q22-08M head by Sandvik Coromant.

**Figure 3 materials-15-03626-f003:**
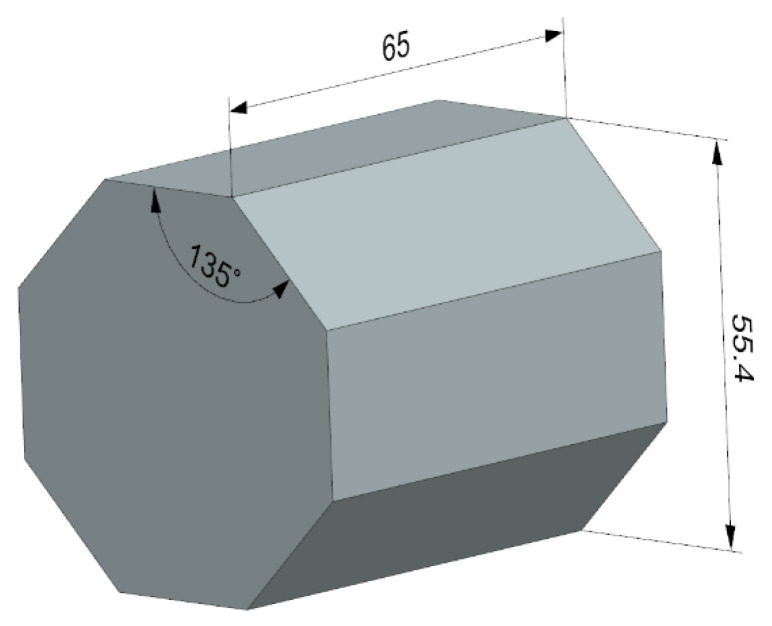
Dimensions of the test sample.

**Figure 4 materials-15-03626-f004:**
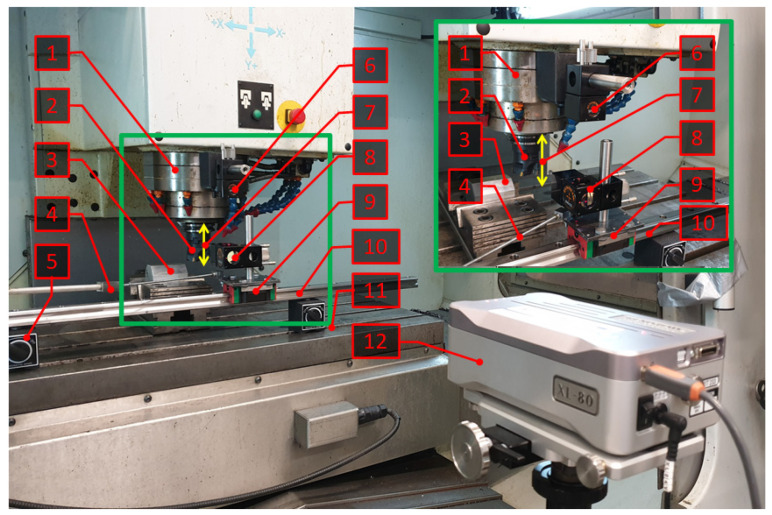
Test bench for measuring relative displacement in a tool-workpiece system. 1—spindle; 2—milling head; 3—workpiece; 4—link; 5—magnetic base; 6—linear reflector; 7—displacement in the tool-workpiece system; 8—linear interferometer rail; 9—block; 10—rail; 11—table; 12—XL 80 laser.

**Figure 5 materials-15-03626-f005:**
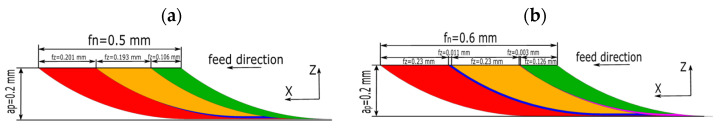
Diagram of the measurement of the inaccuracy of the clamping of the cutting inserts in the body of the milling head: (**a**) for the feed *f_z_* = 0.1, (**b**) for the feed *f_z_* = 0.12.

**Figure 6 materials-15-03626-f006:**
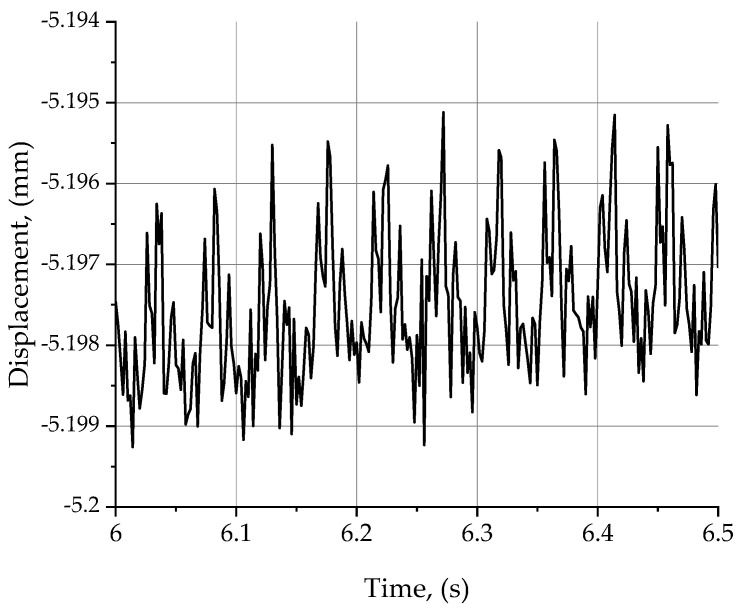
Relative displacement signal plot for *v_c_* = 200 m/min, *f_z_* = 0.1 mm/tooth, *a_p_* = 1 mm, *a_e_* = 24 mm, *z* = 4.

**Figure 7 materials-15-03626-f007:**
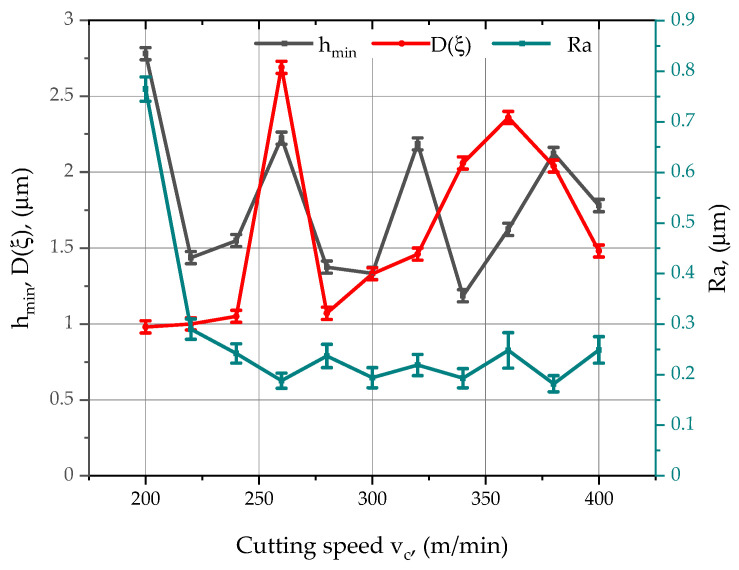
Effect of changing *v_c_* and selected machining conditions on the *Ra* parameter of the surface roughness of aluminium alloy 2017A.

**Figure 8 materials-15-03626-f008:**
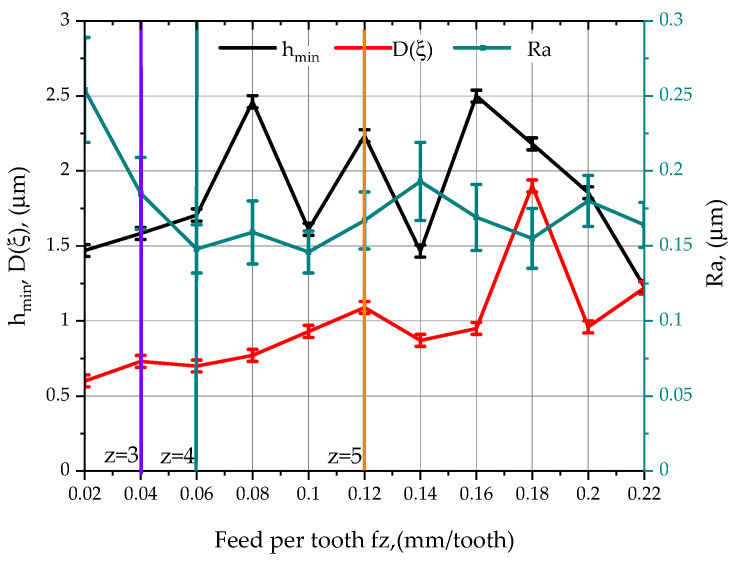
Effect of *f_z_* feed rate changing and selected machining conditions on the *Ra* parameter of the surface roughness of aluminium alloy 2017A.

**Figure 9 materials-15-03626-f009:**
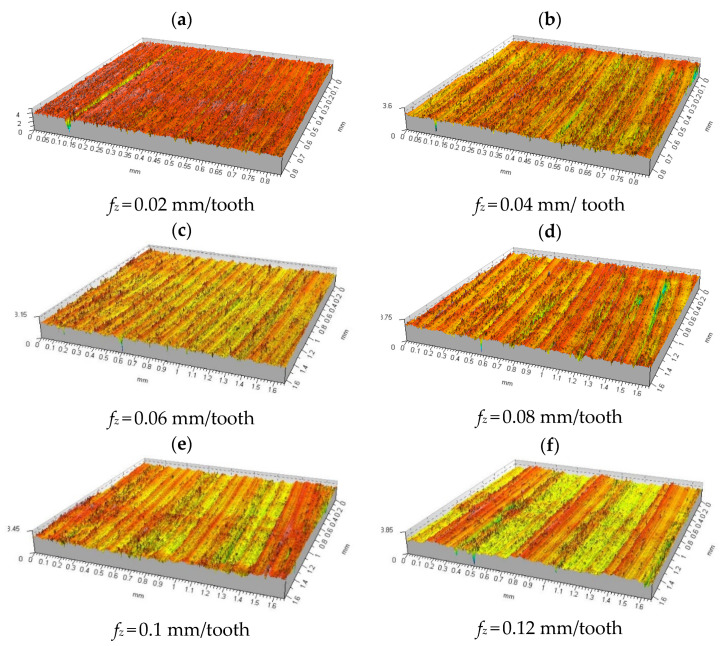
Summary of isometric images of the surface microgeometry of samples made of 2017A alloy machined with variable feed rate. Cutting parameters: *v_c_* = 300 m/min, *n* = 1911 rev./min, *a_p_* = 0.2 mm.

**Figure 10 materials-15-03626-f010:**
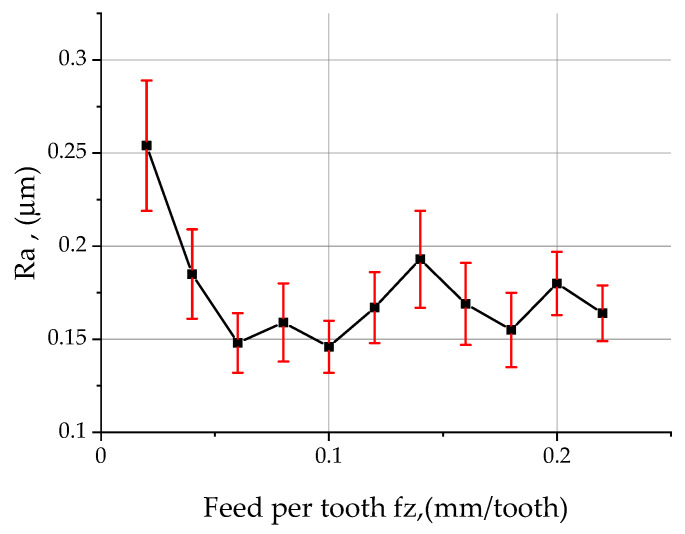
Effect of feed rate on the Ra parameter of surface roughness during face milling of samples made of 2017A alloy.

**Figure 11 materials-15-03626-f011:**
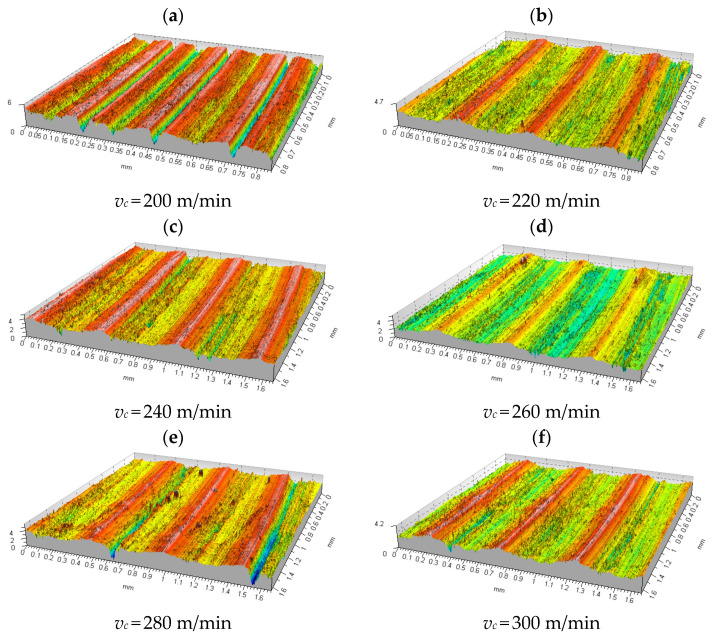
Summary of isometric images of the surface microgeometry of samples made of the 2017A alloy machined at variable cutting speeds. Cutting parameters: *f_z_* = 0.1 mm/tooth, *a_p_* = 0.2 mm.

**Figure 12 materials-15-03626-f012:**
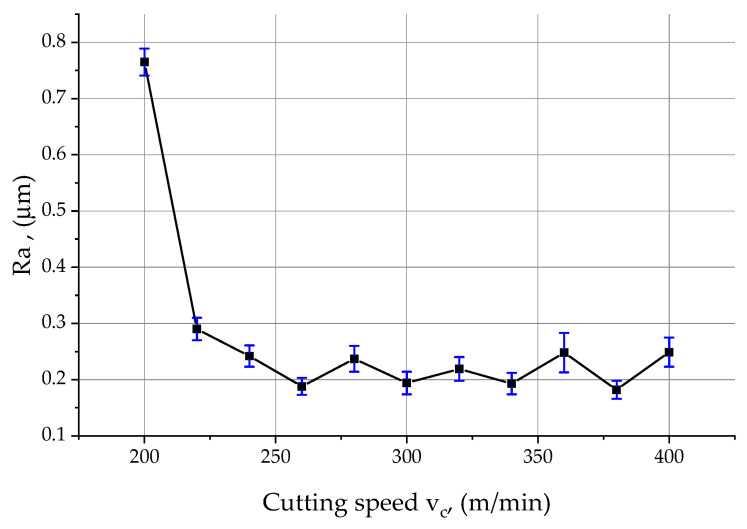
Effect of cutting speed on the Ra parameter of surface roughness during face milling of samples made of 2017A alloy.

**Table 1 materials-15-03626-t001:** Composition of alloying elements of the workpiece material, %.

Cu	Mg	Mn	Si	Fr	Zr + Ti	Zn	Cr	Other	Other in Total	Al
3.5 ÷ 4.5	0.40 ÷ 1.0	0.40 ÷ 0.1	0.20 ÷ 0.8	max. 0.7	max. 0.02	max. 0.02	max. 0.10	max. 0.05	max. 0.15	91.5–95.5

**Table 2 materials-15-03626-t002:** Basic mechanical and physical properties of the workpiece material.

Hardness HV 10	Yield Point R_m_, MPa	Specific Cutting Resistance k_c_ 1.1, N/mm^2^	Density, g/cm³	Young’s Modulus, MPa	Pour Point, °C	Thermal Expansion, µm/mK	Thermal Conductivity, W/mK
106	215 ÷ 295	830	2.79	72,500	645	22.9	134

**Table 3 materials-15-03626-t003:** Errors in clamping the cutting inserts in the body of the milling head and the number of teeth taking an active part in cutting for specified feed rate intervals.

Clamping Error, μm	Cutting Insert Number				
	1	2	3	4	5
Δ*L*	0	−0.72	−9.51	−11.04	−3.25
*2σ*—Δ*L*	6.31	6.36	6.45	6.30	6.30
Δ*R*	−11.33	−4.46	−111.18	0	−117.98
2σ—ΔR	1.16	1.33	23.91	1.31	24.16

**Table 4 materials-15-03626-t004:** Summary of surface roughness parameters of samples machined with variable feed per cutting insert.

Roughness Parameter	Feed Per Tooth *f_z_*, mm/min											Raw Surface
	0.02	0.04	0.06	0.08	0.10	0.12	0.14	0.16	0.18	0.20	0.22	-
Rp *, μm	0.818	0.765	0.552	0.589	0.529	0.643	0.589	0.601	0.544	0.598	0.601	9.610
Rp **, μm	0.137	0.150	0.095	0.114	0.091	0.118	0.098	0.111	0.101	0.103	0.101	0.568
Rv *, μm	1.855	1.220	0.811	0.770	0.666	0.733	0.812	0.782	0.663	0.933	0.681	8.590
Rv **, μm	0.408	0.222	0.137	0.185	0.145	0.122	0.148	0.119	0.201	0.149	0.117	0.293
Rz *, μm	2.673	1.986	1.363	1.359	1.196	1.376	1.401	1.383	1.207	1.531	1.282	18.20
Rz **, μm	0.451	0.290	0.187	0.249	0.190	0.185	0.156	0.193	0.270	0.197	0.180	0.603
Rc *, μm	0.688	0.587	0.471	0.535	0.458	0.496	0.569	0.502	0.485	0.582	0.489	18.00
Rc **, μm	0.102	0.080	0.055	0.086	0.054	0.057	0.091	0.064	0.081	0.080	0.057	0.333
Rt *, μm	3.410	2.300	2.096	2.261	1.871	2.213	2.372	2.120	2.226	2.800	2.033	18.20
Rt **, μm	0.792	0.359	0.302	0.419	0.384	0.351	0.368	0.386	0.662	0.277	0.374	0.603
Ra *, μm	0.254	0.185	0.148	0.159	0.146	0.167	0.193	0.169	0.155	0.180	0.164	4.500
Ra **, μm	0.035	0.024	0.016	0.021	0.014	0.019	0.026	0.022	0.020	0.017	0.015	0.050
Rq *, μm	0.390	0.267	0.207	0.217	0.193	0.226	0.257	0.229	0.207	0.257	0.213	5.230
Rq **, μm	0.064	0.037	0.023	0.034	0.023	0.026	0.036	0.033	0.036	0.025	0.023	0.049
RSm *, mm	0.012	0.011	0.023	0.026	0.025	0.026	0.032	0.027	0.030	0.032	0.031	0.354
RSm **, mm	0.002	0.003	0.003	0.003	0.004	0.003	0.007	0.004	0.003	0.004	0.004	0.001
Rdq *, °	14.244	12.365	4.852	4.605	4.507	4.758	4.663	4.891	4.211	4.440	4.425	7.810
Rdq **, °	1.489	1.574	0.604	0.674	0.641	0.558	0.431	0.661	0.739	0.638	0.635	0.184
Sq, μm	0.384	0.278	0.229	0.281	0.267	0.326	0.444	0.375	0.480	0.495	0.622	5.120
Sp, μm	1.212	1.189	1.115	1.155	1.105	1.378	1.379	1.428	1.503	1.390	1.956	15.50
Sv, μm	3.518	2.227	1.790	2.280	2.103	2.158	2.631	2.269	2.096	2.165	2.082	10.80
Sz, μm	4.730	3.416	2.905	3.434	3.208	3.535	4.010	3.697	3.599	3.555	4.037	26.30
Sa, μm	0.236	0.191	0.156	0.191	0.203	0.250	0.345	0.303	0.397	0.421	0.527	4.560

*—average value, **—standard deviation.

**Table 5 materials-15-03626-t005:** Summary of surface roughness parameters of samples machined with a variable cutting speed.

Roughness Parameter	Cutting Speed *v_c_*, m/min											Raw Surface
	200	220	240	260	280	300	320	340	360	380	400	-
Rp *, μm	1.398	9.610	0.785	0.740	0.869	0.693	0.778	0.708	0.805	0.656	0.786	9.610
Rp **, μm	0.121	0.568	0.093	0.123	0.149	0.100	0.108	0.111	0.135	0.100	0.101	0.568
Rv *, μm	3.250	8.590	1.055	0.719	1.058	0.783	0.938	0.837	0.974	0.698	0.991	8.590
Rv **, μm	0.172	0.293	0.127	0.112	0.207	0.122	0.119	0.114	0.125	0.112	0.190	0.293
Rz *, μm	4.649	18.20	1.840	1.459	1.927	1.476	1.715	1.545	1.780	1.354	1.777	18.20
Rz **, μm	0.224	0.603	0.175	0.191	0.306	0.181	0.174	0.181	0.224	0.177	0.238	0.603
Rc *, μm	2.000	18.00	0.678	0.639	0.765	0.616	0.651	0.593	0.702	0.576	0.699	18.00
Rc **, μm	0.417	0.333	0.061	0.079	0.095	0.065	0.066	0.068	0.091	0.064	0.085	0.333
Rt *, μm	4.872	18.20	2.675	2.247	3.278	2.267	2.446	2.226	2.929	2.002	2.799	18.20
Rt **, μm	0.291	0.603	0.364	0.394	0.480	0.331	0.364	0.348	0.556	0.318	0.460	0.603
Ra *, μm	0.765	4.500	0.242	0.188	0.237	0.194	0.219	0.193	0.248	0.182	0.249	4.500
Ra **, μm	0.024	0.050	0.019	0.015	0.023	0.020	0.021	0.019	0.035	0.016	0.026	0.050
Rq *, μm	0.984	5.230	0.327	0.247	0.332	0.257	0.293	0.256	0.326	0.236	0.330	5.230
Rq **, μm	0.028	0.049	0.027	0.025	0.044	0.029	0.028	0.027	0.045	0.023	0.040	0.049
RSm *, mm	0.045	0.354	0.029	0.027	0.033	0.028	0.027	0.027	0.033	0.028	0.029	0.354
RSm **, mm	0014	0.001	0005	0004	0005	0004	0005	0.004	0.005	0.003	0.005	0.001
Rdq *, °	15.577	7.810	6.009	5.199	5.762	5.170	2.851	5.544	5.657	4.853	5.983	7.810
Rdq **, °	1.820	0.184	0.642	0.656	0.651	0.630	0.720	0.744	0.628	0.703	0.656	0.184
Sq, μm	0.995	5.120	0.590	0.518	0.710	0.539	0.549	0.504	0.633	0.495	0.571	5.120
Sp, μm	1.876	15.50	1.637	2.606	2.099	1.618	1.890	1.662	1.713	1.545	1.896	15.50
Sv, μm	3.806	10.80	2.893	2.114	3.558	2.206	2.573	2.041	3.702	1.801	2.964	10.80
Sz, μm	5.682	26.30	4.530	4.720	5.657	3.823	4.463	3.703	5.416	3.345	4.860	26.30
Sa, μm	0.790	4.560	0.475	0.425	0.535	0.446	0.444	0.412	0.477	0.412	0.453	4.560

*—average value, **—standard deviation.

## Data Availability

All research and analysis results are archived at the Kielce University of Technology, Faculty of Mechanical Engineering, Department of Mechanical Technology and Metrology.

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
