# Peer review of "Influence of the Milling Conditions of Aluminium Alloy 2017A on the Surface Roughness"

_materials, 2022, doi:10.3390/ma15103626_

Round 1

Reviewer 1 Report

The manuscript entitled “Influence of the milling conditions of Aluminum alloy 2017A on the surface roughness “authored by Nowakowski et al. attempts to determine the influence of milling operation on the surface roughness of Aluminum. The topic is relevant. The authors’ effort in performing an experimental work is commendable. However, the article needs significant revision before being considered for publication.

There are some of the points are given for the author’s perusal:

  • Authors advise to add "Aluminum" to Keywords;
  • The Abstract is very concise; it needs to be rewritten again with a few findings and quantitative details;
  • The problem statement and research gap are not clear in the article;
  • Besides it is unreadable the quality of Fig1 should be improved;
  • Authors are directed to add other mechanical properties to table 2, such as Tensile strength, yield strength, Modulus, elongation;
  • What are Rm and Kc, authors directed to identify the terminologies before using the abbreviation;
  • The methodology section does not give enough detail about surface roughness measurements replication;
  • Authors advise considering a deeper literature review and a final discussion relating the findings of the research with previous works;
  • It is not clear how were the cutting parameters (e.g., speed, feed) determined? Preliminary experiments, literature knowledge, experience, etc;
  • For evaluating the machining accuracy, tool wear is very important; I did not see any measurement or investigation about this.  
  • The conclusion section is weak, as it is essentially just bullet points, it should be improved;

Overall, the topic of the article is relevant and interesting and deserves the attention of readers. After the proposed comments were made, the article can be accepted for publication in the journal.

Author Response

Dear Reviewer;

Thank You for all your comments on my article.

Below are my comments on your comments:

Authors advise to add "Aluminum" to Keywords;

The word aluminum was added to the keywords.

The Abstract is very concise; it needs to be rewritten again with a few findings and quantitative details;

As suggested by the reviewer, the abstract has been corrected.

The problem statement and research gap are not clear in the article;

The problem statement and research gap has been corrected.

Besides it is unreadable the quality of Fig1 should be improved;

As suggested by the reviewer, the legibility of Figure 1 has been improved.

Authors are directed to add other mechanical properties to table 2, such as Tensile strength, yield strength, Modulus, elongation;

The quantities mentioned by the reviewer were added in table 2.

What are Rm and Kc, authors directed to identify the terminologies before using the abbreviation;

The meaning of the abbreviations Rm and kc is explained in Table 2.

The methodology section does not give enough detail about surface roughness measurements replication;

The missing information was added to the text at the end of chapter 2.

Authors advise considering a deeper literature review and a final discussion relating the findings of the research with previous works;

Missing information about previous research was added in the introduction. This information was supported by appropriate references to the literature.

It is not clear how were the cutting parameters (e.g., speed, feed) determined? Preliminary experiments, literature knowledge, experience, etc;

The cutting process parameters were adopted on the basis of previous research. Appropriate annotation about it was added in the text.

For evaluating the machining accuracy, tool wear is very important; I did not see any measurement or investigation about this.

Aluminum 2017A is an alloy with low hardness and good machinability. Therefore, in the conducted experiments, no signs of wear of the cutting tool were observed. In addition, new blades were used each time for testing. For this reason, the article ignores the influence of cutting blades wear on the microgeometry of the tested surfaces. Appropriate annotation about it was added in the text.

The conclusion section is weak, as it is essentially just bullet points, it should be improved;

As suggested by the reviewer, the conclusions section has been improved.

Overall, the topic of the article is relevant and interesting and deserves the attention of readers. After the proposed comments were made, the article can be accepted for publication in the journal.

Best regards,

Autor

Reviewer 2 Report

Comments to improve the article are as follows:

  1. Mention the main results and main conclusions in the abstract.
  2. In the Introduction section you did not mention the impact of fixtures. Previous research on fixtures should also be mentioned. Stiffness and compliance of fixture have an impact on surface roughness.
  3. Previous research is numerous. Complete the review of previous research with some more recent manuscripts.
  4. You try to emphasize the innovation of your methodology. What's new in methods and / or experimental research?
  5. Further elaborate on the universality of your methodology.
  6. In addition to mechanical characteristics, it would be good to show the physical and thermal characteristics of the workpiece.
  7. Elaborate on the selection of cutting tools. Show and explain the choice of geometric parameters of the cutting insert.
  8. Analyze and discuss potential errors.
  9. Assess the measurement uncertainty of your results.
  10. Initial surface roughness parameters should be displayed. The obtained results should be compared with the initial parameters of surface roughness.
  11. The discussion of the obtained results must be more intensive. It is not enough to state that something rises or falls with the increase or decrease of something. It is necessary to procedurally explain why this is so. Process physics must be discussed. This is also the biggest drawback of this research.
  12. Also, the values of surface roughness parameters shown in Tables 4 and 5 should be further discussed. Each parameter needs to be further analyzed and discussed.
  13. In the conclusions, mention the limitations of your methodology and future research.

Author Response

Dear Reviewer;

Thank You for all your comments on my article.

Below are my comments on your comments:

  1. Mention the main results and main conclusions in the abstract.

Corrected as suggested by the reviewer

  1. In the Introduction section you did not mention the impact of fixtures. Previous research on fixtures should also be mentioned. Stiffness and compliance of fixture have an impact on surface roughness.

Missing information about the previous research on the dynamic stiffness of the tool, the dynamic stiffness of the machine tool and the effect of the method of clamping the workpiece on the roughness of the machined surface was added in the introduction. This information was supported by appropriate references to the literaturÄ™.

  1. Previous research is numerous. Complete the review of previous research with some more recent manuscripts.

As suggested by the reviewer, the introduction has been corrected and a discussion of research not previously shown there has been added. This information was supported by appropriate references to the literature.

  1. You try to emphasize the innovation of your methodology. What's new in methods and / or experimental research?

The innovativeness of the issues discussed in the article lies in the fact that until now it is believed that only the basic parameters of the technological process affect the roughness of the machined surface. Meanwhile, as proven in this article, many factors influence roughness. Moreover, the innovation is evidenced by the fact that the research was carried out on a special measuring device, the design details of which are protected by patent P.393156 "Machine tool and laser system set for measuring relative displacements". This information was provided at the end of the introduction.

  1. Further elaborate on the universality of your methodology.

The station in question is universal and can be installed on any milling machine, regardless of its kinematic system, manufacturer, year of construction and control method. Appropriate information about this was added in the applications.

  1. In addition to mechanical characteristics, it would be good to show the physical and thermal characteristics of the workpiece.

Added in table 2.

  1. Elaborate on the selection of cutting tools. Show and explain the choice of geometric parameters of the cutting insert.

Added in the text.

  1. Analyze and discuss potential errors.

Added in the text

  1. Assess the measurement uncertainty of your results.

Added in the text.

  1. Initial surface roughness parameters should be displayed. The obtained results should be compared with the initial parameters of surface roughness.

Added in the text and table 4 and 5.

  1. The discussion of the obtained results must be more intensive. It is not enough to state that something rises or falls with the increase or decrease of something. It is necessary to procedurally explain why this is so. Process physics must be discussed. This is also the biggest drawback of this research.

As suggested by the reviewer, the text added a description referring to the physics of the cutting process.

  1. Also, the values of surface roughness parameters shown in Tables 4 and 5 should be further discussed. Each parameter needs to be further analyzed and discussed.

As suggested by the reviewer, the roughness parameters shown in Tables 4 and 5 have been discussed in more detail. The appropriate fragment of the description has been added to the text.

  1. In the conclusions, mention the limitations of your methodology and future research.

Added in the conclusions.

Best regards,

Autor

Reviewer 3 Report

The paper deals with the Influence of the milling conditions of aluminium alloy 2017A on the surface roughness. 
According to the reviewer, the paper is worth publishing at Materials Journal, but corrections are needed.
While the authors have made considerable research effort, 
the presentation of the paper and the results must be proven.
Additionally make the following corrections to the manuscript:

Comment 1
The authors should explain the difference between the tooth and blade.
Feed per tooth fz (mm/tooth) vs fz (mm/blade)?
If the same, the authors should replace at all paper the word "blade" with the tooth.

Comment 2
Line 69
Oliveira et al [14] found
The authors should replace (insert a .)
Oliveira et al. [14] found

Comment 3
Lines 75 - 78 + Figure 1
In Figure 1 there are words that are not written in English.
The authors should explain the difference: 
machining parameters (seven variables in the text vs 8 in the Figure), 
tool properties (four variables in the text vs 9 in the Figure), 
workpiece material (three variables in the text vs 6 in the Figure) 
and phenomena accompanying the cutting process (five variables in the text vs 8 in the Figure). 

Comment 3
Line 86
et al [20] point out
The authors should replace (insert a .)
et al. [20] point out 

Comment 4
Table 1
The authors should add the whole elements (sum 100%).
The authors should explain if the values (Table 1 and 2) are for their experiments or for the supplier.

Comment 5
Line 124
cross-section (Figure 3.). This
The authors should replace (delete a .)
cross-section (Figure 3). This

Comment 6
The authors should insert in the Figure 5 more details (dimensions for each color).

Comment 7
Lines 240 - 253
The authors should format the text according to the journal's instructions (full alignment).

Comment 8
The authors should explain why if the Feed per tooth fz,(mm/tooth) increased, the Ra decreased (Table 4 and Figure)? 

Comment 9
Line 413
(fig. 12). In
The authors should replace
(Figure 12). In

Line 420
(Tab. 5). A
The authors should replace
(Table 5). A

Line 360
(Tab. 4). The a
The authors should replace
(Table 4). The a

Line 281
It is not so good to use the word "we".

Comment 10
Lines 280 - 286
The authors should check if the values are really 0mm2.

Comment 11
The authors should explain the difference between the fz (mm/tooth), fn (mm/rev.), ft (mm/min)

Comment 12
The authors should format the References according to the journal 's instructions (Volume Italics).
References should be described as follows, depending on the type of work:
Journal Articles:
1. Author 1, A.B.; Author 2, C.D. Title of the article. Abbreviated Journal Name Year, Volume, page range. 

Author Response

Dear Reviewer;

Thank You for all your comments on my article.

Below are comments on Your suggestions:

Comment 1

The authors should explain the difference between the tooth and blade.

Feed per tooth fz (mm/tooth) vs fz (mm/blade)?

If the same, the authors should replace at all paper the word "blade" with the tooth.

Corrected as suggested by the Reviewer.

Comment 2

Line 69

Oliveira et al [14] found

The authors should replace (insert a .)

Oliveira et al. [14] found

Corrected as suggested by the Reviewer.

Comment 3

Lines 75 - 78 + Figure 1

In Figure 1 there are words that are not written in English.

The authors should explain the difference:

machining parameters (seven variables in the text vs 8 in the Figure), tool properties (four variables in the text vs 9 in the Figure), workpiece material (three variables in the text vs 6 in the Figure) and phenomena accompanying the cutting process (five variables in the text vs 8 in the Figure).

Corrected as suggested by the Reviewer.

Comment 3

Line 86

et al [20] point out

The authors should replace (insert a .)

et al. [20] point out

Corrected as suggested by the Reviewer.

Comment 4

Table 1

The authors should add the whole elements (sum 100%).

The authors should explain if the values (Table 1 and 2) are for their experiments or for the supplier.

Corrected as suggested by the Reviewer.

Comment 5

Line 124

cross-section (Figure 3.). This

The authors should replace (delete a “.”)

cross-section (Figure 3). This

Corrected as suggested by the Reviewer.

Comment 6

The authors should insert in the Figure 5 more details (dimensions for each color).

Corrected as suggested by the Reviewer.

Comment 7

Lines 240 - 253

The authors should format the text according to the journal's instructions (full alignment).

Corrected as suggested by the Reviewer.

Comment 8

The authors should explain why if the Feed per tooth fz,(mm/tooth) increased, the Ra decreased (Table 4 and Figure)?

Corrected as suggested by the Reviewer.

Comment 9

Line 413

(fig. 12). In

The authors should replace

(Figure 12). In

Corrected as suggested by the Reviewer.

Line 420

(Tab. 5). A

The authors should replace

(Table 5). A

Corrected as suggested by the Reviewer.

Line 360

(Tab. 4). The a

The authors should replace

(Table 4). The a

Corrected as suggested by the Reviewer.

Line 281

It is not so good to use the word "we".

Corrected as suggested by the Reviewer.

Comment 10

Lines 280 - 286

The authors should check if the values are really 0mm2.

Corrected as suggested by the Reviewer.

Comment 11

The authors should explain the difference between the fz (mm/tooth), fn (mm/rev.), ft (mm/min)

Corrected as suggested by the Reviewer.

Comment 12

The authors should format the References according to the journal 's instructions (Volume Italics).

References should be described as follows, depending on the type of work:

Journal Articles:

  1. Author 1, A.B.; Author 2, C.D. Title of the article. Abbreviated Journal Name Year, Volume, page range.

Corrected as suggested by the Reviewer.

Best regards,

Autor

Reviewer 4 Report

In general, the article is written well, and the research is presented at a high level. The Methodology section describes the equipment used and the operating modes of this equipment. The presented results of the performed experiments are of scientific and practical interest. The review of the already conducted studies is also performed at a proficient level. In general, improving the accuracy of processing aluminum alloy workpieces with a multi-blade tool is an urgent and interesting task, both from the point of view of science, and of production. 
The following improvements can be made for the article to be better perceived:
The confidence probability is not indicated in Figures 7 and 8. 
In the second point of conclusions, you write that
• Characteristic cutting speed ranges of 200 - 240 m/min and 280 m/min were deter-438 mined where the standard deviation of relative displacement had the lowest value.
give this minimum value here in the conclusion.
It would be also good to add quantitative data obtained in this work to the conclusions.
And about the following part of conclusion:
• The value of the standard deviation of the relative displacements shows an increasing trend as the feed per blade increases. However, there are characteristic boundaries responsible for increasing the proportion of inserts involved in the material removal process that reverse or significantly reduce this trend.
It would be extremely useful to reflect information on what these boundaries depend on and in which case they reverse this trend.

Author Response

Dear Reviewer;

Thank You for all your comments on my article.

Below are comments on Your suggestions:

The confidence probability is not indicated in Figures 7 and 8.

As suggested by the reviewer, the missing information was supplemented in Figures 7 and 8.

In the second point of conclusions, you write that

  • Characteristic cutting speed ranges of 200 - 240 m/min and 280 m/min were deter-438 mined where the standard deviation of relative displacement had the lowest value.

give this minimum value here in the conclusion.

Corrected as suggested by the reviewer.

It would be also good to add quantitative data obtained in this work to the conclusions.

As suggested by the reviewer, the quantitative data obtained in the work were quoted in the applications.

And about the following part of conclusion:

  • The value of the standard deviation of the relative displacements shows an increasing trend as the feed per blade increases. However, there are characteristic boundaries responsible for increasing the proportion of inserts involved in the material removal process that reverse or significantly reduce this trend.

It would be extremely useful to reflect information on what these boundaries depend on and in which case they reverse this trend.

Corrected as suggested by the reviewer - supplemented in conclusion 7

Best regards,

Autor

Round 2

Reviewer 1 Report

Article can be published after address all comments. 

Author Response

As suggested by the reviewer, all corrections were introduced in the article and the comments of reviewers were answered.

Reviewer 2 Report

The manuscript has been corrected.

Author Response

As suggested by the reviewer, the text of the article was checked and corrected.

Reviewer 3 Report

Comment 1
Line 87
The paper [19
Text is missing.

Comment 2
Line 101
which was also described by Dzyura et al. [ 
Text is missing.
Extended text editing

Comment 3
The authors must comment the ref. [26] and [28] in the paper.

Comment 4
The Figure 4 must be accompanied on the same page as the Figure's title.
The Figure 8 must be accompanied on the same page as the Figure's title.

Comment 5
Line 206
aluminium 2017A
The authors should replace
Aluminium 2017A

Comment 6
Tab.
The authors should replace
Table

Comment 7
Table 4
Ra = 4.500 μm for 220 cutting speed m/min
The authors must check if the value is correct.

Author Response

Line 87
The paper [19
Text is missing.

The missing text fragment was completed.

Comment 2
Line 101
which was also described by Dzyura et al. [ 
Text is missing.
Extended text editing

The missing text fragment was completed.

Comment 3
The authors must comment the ref. [26] and [28] in the paper.

Ref. [26] was quoted in the text that was accidentally deleted (Comment 2).

Nr ref. [28] cytat był niewidoczny z powodu błędu zapisu. Poprawiony cytat znajduje się w wierszu 108.

Comment 4
The Figure 4 must be accompanied on the same page as the Figure's title.
The Figure 8 must be accompanied on the same page as the Figure's title.

As suggested by the reviewer, the drawings were moved.

Comment 5
Line 206
aluminium 2017A
The authors should replace
Aluminium 2017A

Corrected as suggested by the reviewer.

Comment 6
Tab.
The authors should replace
Table

Corrected as suggested by the reviewer.

Comment 7
Table 4
Ra = 4.500 μm for 220 cutting speed m/min
The authors must check if the value is correct.

The Ra parameter value is correct and consistent with the test results. The obtained surface geometry is the result of the coarse face milling process on a conventional machine.